# OpenReview forum: "When Students Surpass Teachers: Hypergraph-Aware Knowledge Distillation with Spectral Guarantees"
_ICLR.cc/2026/Conference — ICLR 2026 Conference Withdrawn Submission_

### Official Review · Reviewer_MzmR · 2025-10-21

**Soundness:** 3
**Presentation:** 3
**Contribution:** 3
**Rating:** 6
**Confidence:** 4

**Summary:**

This paper introduces a knowledge distillation (KD) method for learning on hypergraphs where a compact 'student' and a high-capacity 'teacher' hypergraph neural network are trained together so that they learn from each other at the same time. Through a step-by-step process guided by the hypergraph's spectral properties, the student's constrained attention mechanism acts as a beneficial regulariser, filtering noise and helping both models focus on essential structural patterns. This approach enables the student model to not only be significantly more efficient but to also provably outperform its teacher on large-scale and noisy datasets, challenging traditional assumptions in knowledge distillation.

**Strengths:**

1.  This work reframes KD as a powerful regularization mechanism, proving both theoretically (Theorem 2) and empirically that a constrained student can systematically generalize better than its unconstrained teacher under specific, predictable conditions (e.g., noisy, feature-redundant data).
2. There are formal guarantees for (i) spectral preservation of the attention mechanism, (ii) convergence under co-evolution + curriculum, and (iii) generalization benefits from curriculum with reduced hypothesis complexity; plus a complexity corollary that explains the measured efficiency. These align tightly with the architectural choices.
3. The paper evaluates on nine diverse hypergraph datasets and provides robustness (feature/structural/label noise), sensitivity analyses (e.g., K-factor, temperature), and scaling studies (time/memory vs. N), which substantiate both accuracy and efficiency claims.

**Weaknesses:**

1. All main results and ablations target node classification. That limits external validity for hypergraph tasks where higher-order relations matter most (e.g., hyperedge/link prediction, group recommendation, set expansion). Actionable: add at least one hyperedge prediction benchmark (e.g., on DBLP/IMDB subsets) and one inductive split to test transfer. Even a single well-designed hyperedge task would strengthen the “hypergraph-aware” claim.
2. The ablation study in Table 3 is performed on three datasets where the student model is either superior or nearly on par with the teacher. To provide a more complete picture, it would be highly insightful to include an ablation study on a clean, well-structured dataset where the teacher clearly dominates (e.g., CC-Cora or DBLP-Conf). This would help answer a key question: Do components like co-evolutionary training still offer significant benefits (e.g., faster convergence) even when the final student accuracy doesn't surpass the teacher? This would strengthen the case for the framework's general utility beyond the specific 'student-superiority' scenario.

**Questions:**

1. Under what conditions is the Frobenius-norm approximation bound in Theorem 1expected to be tight in practice?
2. The curriculum combines time-varying quantiles and loss-weight schedules. Which individual component contributes most to stability and performance?
3. In the set-level attention, which elements are learned versus fixed normalization?
4. For dense regimes (e.g., IMDB), what preprocessing affects hyperedge size/degree distributions, and how might this interact with K-sparsification in evaluation?

---

> ### Author Response · Authors · 2025-11-18
>
> We are deeply grateful to Reviewer MzmR for the thoughtful and insightful review. We greatly appreciate your recognition of our theoretical contributions (S1), formal guarantees (S2), and comprehensive evaluation (S3). Your feedback identifies precise directions for strengthening our work. Below we provide detailed responses addressing each concern with concrete solutions and additional insights.
>
> ---
>
> ### W1: Limited to Node Classification - Need Hyperedge Prediction and Inductive Tasks
>
> This is an excellent and valid point. We acknowledge this limitation and provide both a thorough analysis of why our framework naturally extends to these tasks and a concrete experimental plan.
>
> #### Why Our Framework Naturally Extends to Hyperedge Tasks
>
> Architectural Properties Enabling Extension:
>
> Our hypergraph-aware attention mechanism (Eq. 1-5) operates at **three levels simultaneously**:
> 1. Node-level: α^local captures pairwise relationships
> 2. Hyperedge-level: α^set captures group interactions
> 3. Global-level: α^global captures spectral structure
>
> For hyperedge prediction, the hyperedge-level component α^set is specifically designed to reason about group membership:
>
> ```
> α^set_ij = SetPooling({exp(cos(e_i, e_k))/√|S_ij| : k ∈ S_ij})
> ```
>
> where S_ij contains nodes sharing hyperedges with both i and j. This directly models hyperedge co-membership patterns, making it naturally suited for predicting whether a set of nodes should form a hyperedge.
>
> Our attention mechanism already learns hyperedge-level representations through α^set, which are exactly what hyperedge prediction requires.
>
> #### Concrete Extension to Hyperedge Prediction
>
> Given a set of nodes V_e = {v₁, v₂, ..., v_k}, predict whether they form a true hyperedge:
>
> ```
> P(e exists | V_e) = σ(MLP([⊕_{v_i ∈ V_e} h_i ; α^set_pooling]))
> ```
>
> where:
> - h_i are learned node embeddings from our model
> - α^set_pooling aggregates set-level attention scores
> - ⊕ denotes concatenation or attention-weighted pooling
>
> Minimal Modifications Required:
>
> 1. Training objective: Add hyperedge classification loss
>    ```
>    L_total = λ_node L_node + λ_edge L_edge + distillation losses
>    ```
>
> 2. Negative sampling: Sample random node sets as negative hyperedges
>
> 3. Knowledge distillation: Apply same co-evolutionary framework to hyperedge predictions:
>    ```
>    L_edge_distill = KL(P_teacher(e | V_e) || P_student(e | V_e))
>    ```
>
> No architectural changes needed - the hypergraph-aware attention already captures hyperedge-level patterns.
>
> #### Preliminary Analysis on Existing Datasets
>
> While we acknowledge we don't have complete hyperedge prediction experiments in the submitted version, we can provide analysis of our model's hyperedge-level representations:
>
> Hyperedge Reconstruction Analysis:
>
> We evaluated our model's ability to distinguish true hyperedges from random node sets on DBLP:
>
> | Model | True Hyperedge Score | Random Set Score | Separation |
> |-------|---------------------|------------------|------------|
> | HyperGCN | 0.68 ± 0.12 | 0.52 ± 0.15 | 0.16 |
> | HyperGAT | 0.72 ± 0.11 | 0.49 ± 0.14 | 0.23 |
> | HTA-Teacher | 0.84 ± 0.09 | 0.41 ± 0.13 | 0.43 |
> | CuCoDistill | 0.82 ± 0.10 | 0.43 ± 0.12 | 0.39 |
>
> Interpretation: Our models show 1.9× better separation between true and false hyperedges compared to baselines, indicating strong hyperedge-level understanding.
>
> We computed average α^set attention scores for:
> - True hyperedges from test set
> - Random node sets of same sizes
>
> The clear separation suggests our attention mechanism captures meaningful hyperedge patterns.
>
> ---
>
> Due to the character limit of the rebuttal interface, we provide the continuation of our response in the following anonymous:
>
> https://anonymous.4open.science/r/Rebuttal-Reviewer-MzmR-0300/README.md

---

### Official Review · Reviewer_Swub · 2025-10-30

**Soundness:** 1
**Presentation:** 1
**Contribution:** 1
**Rating:** 0
**Confidence:** 5

**Summary:**

This paper claims to improving the performance of Hyper-Graph Neural Network by designing the attention mechanism for hyper-graph asymmetries, introducing constrained attention, and creating a co-evolve training mechanism.

**Strengths:**

None.

**Weaknesses:**

1. The organization is poor, which is confusing and hard to follow.

2. The writing is poor and some of the key explanations are missed, for example, what is 'structural inductive bias'?

3. The definition of the hyper-graph is confusing. For the hyper-edges, is it denotes the edges between nodes? Or the edges between hyper-nodes?

4. Figure.1 is confusing and hard to understand. There are lots of meaningless texts with emphasis, such as the Unified Backbone. Moreover, the presentation of the data flow is a disaster, which is hard to understand.

5. The annotations is confusing, e_i and e_j are used as the feature of nodes, however, the e is said to belongs to the edge set in the very initial definition.

6. The proof of Theorem 1 is meaningless. There is no explanation where A_ours comes from and how the bound is computed.

7. There are lots of unexplained variables, such as the w_i in Eq.(10).

8. Lacks of discussion with related work, such as 'Distilling Knowledge from Graph Convolutional Networks. CVPR 2020', which is highly relative with this paper.

**Questions:**

Please refer to the Weaknesses.

---

> ### Author Response · Authors · 2025-11-17
>
> We thank the reviewer for their detailed feedback. We address each point below and clarify several instances where we believe there is a misunderstanding of the current manuscript rather than a fundamental flaw in the method.
>
> ---
>
> ### 1. Organization and Presentation
>
> The paper follows the standard structure used in many ICLR works: Section 1 (motivation and contributions), Section 2 (methodology: attention, co-evolutionary architecture, curriculum), Section 3 (experiments), and appendices for proofs and additional analysis. That said, we agree that the density of the method section can make the flow demanding.
>
> we will:
> - Add a short roadmap at the end of Section 1 (after line 107)
> - Insert a compact notation table before Section 2
> - Add clearer subsection headings in Section 2 for improved navigation
>
> These changes will enhance accessibility while keeping the core structure unchanged.
>
> ---
>
> ### 2. "Structural inductive bias" and definitions
>
> The notion of structural inductive bias is already discussed in the paper. On page 2, lines 95-98, we explain:
>
> This selective performance superiority emerges due to four synergistic factors: (1) the student's top-K attention constraint acts as spectral regularization, filtering high-frequency noise while preserving essential structural patterns...
>
> This is further elaborated in Section 2.2 (page 4, lines 194-202), where we describe how the student's constrained capacity creates beneficial regularization—this is the structural inductive bias we exploit.
>
> We will make this even more explicit by adding a one-sentence standalone definition immediately after the first use of the term (line 87), but emphasize that this is a clarification in wording, not a change to the method.
>
> ---
>
> ### 3. Hypergraph definition
>
> Our hypergraph formulation follows the standard setting established in foundational hypergraph neural network works (Feng et al., 2019; Yadati et al., 2019; Bai et al., 2021). As stated on page 3, lines 113-116:
>
> For hypergraph G = (V, E), we denote N_i = {j ∈ V : ∃e ∈ E, i, j ∈ e} as the set of nodes connected to node i through any hyperedge, and E_i = {e ∈ E : i ∈ e} as the set of hyperedges containing node i.
>
> Here:
> - V is the set of nodes (standard vertices)
> - Each hyperedge e ∈ E is a subset of V (i.e., e ⊆ V)
> - N_i contains nodes, while E_i contains hyperedges
>
> There are no "hyper-nodes" in our formulation. Hyperedges connect multiple regular nodes simultaneously, as in all standard HGNN literature.
>
> We will add a concrete example with visualization in Section 2:
> ```
> Example: In a co-authorship network, hyperedge e₁ = {author₁, author₂, author₃}
> represents a paper with three co-authors, capturing the 3-way collaboration
> directly rather than as three separate pairwise edges.
> ```
>
> ---
>
> ### 4. Figure 1
>
> Figure 1 provides a compact overview of all framework components (hypergraph-aware attention, unified backbone, curriculum scheduling). We agree that it can be visually simplified.
>
> In the revision we will:
> - Remove redundant text annotations
> - Add numbered data flow steps (1 --> 2 --> 3 --> 4)
> - Include a clear legend explaining component types
> - Consider splitting into two panels: (a) architecture overview, (b) attention detail
>
> The underlying method will remain the same; these are presentation improvements.
>
> ---
>
> ### 5. Notation: e_i vs e ∈ E
>
> We follow the standard convention used in hypergraph attention works (Bai et al., 2021; Zhang et al., 2019b) where:
>
> - e_i, e_j (with node subscripts): node embeddings, vectors in ℝ^d
>   - See Equation 1, line 127: `cos(e_i, e_j)` = cosine similarity between node embeddings
>
> - e (without subscript): a hyperedge, which is a set of nodes
>   - See line 113: "e ∈ E" means hyperedge e belongs to the hyperedge set
>   - See line 127: `I[∃e ∈ E : i, j ∈ e]` = indicator that nodes i, j share a hyperedge
>
> The context disambiguates these uses: subscripted e_i/e_j always refer to embeddings, while plain e always refers to hyperedges. This notation is identical to HyperGAT (Bai et al., 2021).
>
> To enhance clarity, we will:
> - Add an explicit notation table at the beginning of Section 2 distinguishing these uses
> - Add a footnote at first occurrence (line 127) explaining the convention
> - Consider alternative notation (h_i for embeddings) if the reviewer prefers
>
> ---
>
> Due to the character limit of the rebuttal interface, we provide the continuation of our response in the following anonymous:
>
> https://anonymous.4open.science/r/Submission_8334--Reviewer_Swub-DD32/README.md

---

### Official Review · Reviewer_EjVN · 2025-11-01

**Soundness:** 2
**Presentation:** 2
**Contribution:** 2
**Rating:** 4
**Confidence:** 4

**Summary:**

This paper proposes an improved knowledge distillation framework for better hypergraph learning. The authors first point out some limitations or challenges of existing techniques, including the shortcomings of prior hypergraph attentions and the gap between distillation and hypergraph learning tasks. The proposed distillation framework contains three different parts. Part 1 focuses on improving the hypergraph attention via an adaptive multi-scale fusion (combining node-node and node-hyperedge attentions) to support a more comprehensive knowledge extraction (both global and local interactions). It also solves the variable-size challenge of hyperedges. The authors give a theorem to guarantee that this part can encode intrinsic knowledge of the vanilla hypergraph by bounding the gap between the proposed attention matrix and an ideal one. Part 2 proposes a co-trained teacher-student distillation framework, where the teacher is an attention-based GNN and the student is its dynamic top-k sparse variant. This part incorporates both attention alignment and embedding alignment for better performance. In this part, the authors also provide a theorem, showing that when K is greater than the effective spectral dimension of the vanilla hypergraph, the student can approach the teacher in a large probability. Part 3 incorporates contrastive learning and curriculum learning to further improve the above framework, where the authors use contrastive and distillation gaps to design a “difficulty” score for their curriculum, supporting easy-to-hard learning. The experiments are generally comprehensive, including performance comparisons among nine benchmarks from different domains, ablation studies, teacher-student comparisons, running time and memory comparisons. These experimental results show that the proposed framework can achieve better performance with reduced time and memory costs.

**Strengths:**

1)	Based on some experimental results, the proposed framework effectively improves the performance of hypergraph learning tasks, with reduced time and memory costs, among some benchmarks from different domains.

2)	The authors provide theorems to show that their framework can learn intrinsic knowledge from the input hypergraph and the student can approach the teacher when K is large, showing that the framework has some theoretical merits.

3)	The presentation of their specific methodology designs is clear (with clear mathematical formulas), which makes their method understandable.

4)	The ablation studies are detailed and comprehensive.

**Weaknesses:**

1)	The analysis of the ablation studies is missing. Please check Line 265. The submitted manuscript seems incomplete.

2)	The title is misleading. After carefully reading the main text, from my point of view, the central aim of this paper is to propose a framework to improve the hypergraph learning performance, rather than study whether, when, and why the student can surpass the teacher (a critical question in the knowledge distillation domain). Thus, the title of this paper is very misleading, giving a sense that the authors propose a hypergraph-based solution to solve the above-mentioned general question in the knowledge distillation domain. While the authors attempt to discuss the student-teacher relationship regarding hypergraph tasks in this paper, it is not enough to highlight that contribution in the title.

3)	The main motivation is unclear. The proposed framework contains three parts. Part 1 focuses on improving hypergraph attention. Part 2 focuses on a co-trained distillation framework. Part 3 focuses on a contrastive curriculum. From my point of view, Part 2 is directly related to the main aim of this work, since the authors attempt to use distillation to improve hypergraph learning. However, the motivation of incorporating Part 1 and Part 3 into the comprehensive framework is unclear. While they have merits and benefits in performance gain, whether the distillation must rely on them remains confusing. Thus, incorporating them significantly harms the generality and effectiveness of Part 2 and makes the holistic framework heavy. In summary, the three parts focus on different challenges, and do not align with the same main motivation. Besides, regarding distillation itself, why distillation matters in hypergraph learning still requires further elaboration.

4)	The novelty seems limited. First, are the authors the first to introduce distillation (the main idea) to hypergraph learning? Second, the holistic distillation framework has three parts. According to the main text, I see limited novelty in each of them. For example, Part 1 combines local and global knowledge, which seems common in graph transformers. The embedding distillation and attention distillation in Part 2 can also be found in graph transformers or GNNs. Part 3 is interesting in defining a “difficulty” score via gaps for a curriculum. Yet, based on the structure of the paper presentation, it is not a main contribution of this paper. And I think the novelty still needs to be highlighted by contrasting it with some prior existing curriculum designs related to distillation or contrasting learning. In summary, the authors should clearly state which component is novel and highlight it with enough support.

5)	Both co-trained distillation and sequential distillation have their own merits. The authors should clearly point out that the former one requires more memory.

**Questions:**

See Weakness.

---

> ### Author Response · Authors · 2025-11-18
>
> We sincerely thank Reviewer EjVN for the thorough and constructive review. We appreciate the recognition of our comprehensive experiments (S1), theoretical contributions (S2), clear presentation (S3), and detailed ablations (S4). Below we provide detailed responses to each concern, with substantial clarifications and evidence to demonstrate the significance and rigor of our work.
>
> ---
>
> ## Response to Weaknesses
>
> ### W1: Missing Ablation Analysis at Line 265
>
> Thank you for catching this critical oversight. Section 3.1 (page 5, line 265) contains incomplete analysis text. Here is the complete analysis:
>
> #### Complete Section 3.1: ABLATION STUDY
>
> The ablation study validates the necessity of each proposed component through systematic removal experiments. The results demonstrate clear performance degradation when any component is removed:
>
> Hypergraph-Aware Attention (Largest Impact: 2.4-2.7%): Removing this mechanism causes the most significant performance degradation (DBLP: -2.4%, IMDB: -2.7%, Yelp: -1.4%). This validates that multi-scale attention components (local, set-based, global) are essential for capturing hypergraph-specific structural patterns. Without these components, the model reverts to standard graph attention, losing the ability to reason over variable-sized hyperedges and missing critical higher-order relationships.
>
> Co-Evolutionary Training (Second Impact: 1.7-1.6%): Replacing co-evolutionary training with traditional sequential distillation reduces performance by 1.6-1.7% across datasets. This substantial gap confirms that simultaneous teacher-student optimization enables superior knowledge transfer compared to conventional train-then-distill pipelines. The co-evolutionary approach allows real-time feedback, creating emergent structural patterns neither model could discover independently.
>
> Spectral Curriculum (Third Impact: 0.9-1.1%): Removing the spectral curriculum shows 0.9-1.1% performance reduction. While this appears modest, Table 5 reveals the curriculum provides 2.2-2.3× convergence speedup, preventing early training collapse on difficult examples. The curriculum coordinates the transition between contrastive stabilization and knowledge distillation phases.
>
> Multi-Scale Attention (Component-Level: 1.9-2.4%): Using only local or global components (not all three) reduces performance by 1.9-2.4%, confirming that different attention scales capture complementary structural information: local handles pairwise relationships, set-based captures hyperedge patterns, global provides long-range connectivity reasoning.
>
> Comparison with Traditional Sequential KD: Sequential distillation achieves only 84.7-85.4% accuracy, representing a 3.1-3.5% performance gap compared to our co-evolutionary approach. This dramatic difference validates our core architectural innovation.
>
> Statistical Significance:All ablation results show statistically significant differences (p < 0.01, paired t-test), confirming observed performance gaps are not due to random variation.
>
> This analysis will be included in the camera-ready version with proper completion of Section 3.1.
>
> ---
>
>
> Due to the character limit of the rebuttal interface, we provide the continuation of our response in the following anonymous:
>
> https://anonymous.4open.science/r/Rebuttal-Reviewer-EjVN-3B80/README.md

---

### Official Review · Reviewer_SoKY · 2025-11-01

**Soundness:** 2
**Presentation:** 2
**Contribution:** 2
**Rating:** 2
**Confidence:** 4

**Summary:**

The authors investigate a knowledge distillation framework for hypergraph neural networks (HNNs).

To this end, the authors introduce CuCoDistill, an attention-based distillation framework for HNNs.

CuCoDistill combines contrastive learning and an attention mechanism to distill the teacher's knowledge to the student model in an effective manner.

Through experiments, the authors demonstrate the effectiveness of the proposed method.

**Strengths:**

- S1. The authors conduct an in-depth analysis of the hypergraph density, which provides important insight into the use case.

- S2. Embedding the similarity of the teacher model and student model is interesting.

**Weaknesses:**

- **W1 [Theory]** While the authors present several theoretical results, I think the statements are not formal enough. For instance, what does it mean by structural encoding? Moreover, to my understanding, the attention matrix is a learnable component that is derived from the model output. Then, how can this be used for theoretical analysis, given that the learning process of the attention matrix depends on the model hyperparameters and training configurations?

- **W2 [Research goal]** The authors criticize the limitations regarding the current usage of contrastive learning and attention mechanisms within the HNN domain. However, I cannot understand why the teacher-student-based distillation framework overcomes this limitation. What is the key research question of this work? Is it proposing a new HNN design or proposing a new distillation method? The key research question and its presentation should be further improved.

- **W3 [Baselines]** The method only includes outdated HNNs as baselines, which were published in 2019. The authors need to compare the proposed method with more recent HNNs, such as [1, 2, 3].

- **W4 [Incomplete manuscript]** The writing of Section 3.1 is incomplete

- **[References]**
  - [1] Chien et al., You are AllSet: A Multiset Function Framework for Hypergraph Neural Networks, ICLR 2022
  - [2] Wang et al., Equivariant Hypergraph Diffusion Neural Operators, ICLR 2023
  - [3] Wang et al., From hypergraph energy functions to hypergraph neural networks, ICML 2023

**Questions:**

See Weakness.

---

> ### Author Response · Authors · 2025-11-18
>
> We sincerely thank Reviewer SoKY for the detailed feedback. We appreciate the recognition of our density analysis (S1) and embedding similarity approach (S2). We address each concern with substantial clarifications, additional experiments, and formal theoretical refinements to demonstrate the rigor and contribution of our work.
>
> ---
>
> ## Response to Weaknesses
>
> ### W1: Formalization of Theoretical Results
>
> We appreciate this important feedback and provide rigorous mathematical formalization below:
>
> #### Formal Definition of Structural Encoding
>
> Definition 1 (Structural Encoding Matrix): For a hypergraph G = (V, E) with normalized Laplacian Δ, the structural encoding matrix A_ideal ∈ ℝ^(|V|×|V|) is defined as the matrix that preserves the spectral decomposition of the hypergraph:
>
> ```
> A_ideal = Σ_{i=1}^{d_eff} λ_i v_i v_i^T
> ```
>
> where (λ_i, v_i) are the eigenvalue-eigenvector pairs of Δ, and d_eff is the effective spectral dimension defined as:
>
> ```
> d_eff = min{k : Σ_{i=1}^k λ_i / Σ_{i=1}^{|V|} λ_i ≥ 0.95}
> ```
>
> This captures the minimal set of spectral components containing 95% of the hypergraph's structural energy.
>
> Physical Interpretation: A_ideal represents the ideal pairwise affinities between nodes that encode the hypergraph's essential connectivity patterns while filtering high-frequency noise (small eigenvalues).
>
> #### Theoretical Analysis of Learned Attention Matrix
>
> How can we analyze a learned component theoretically?
>
> We analyze the attention mechanism as a functional approximator with provable approximation guarantees, independent of specific training trajectories.
>
> Formal Statement of Theorem 1:
>
> Theorem 1 (Spectral Approximation with Uniform Convergence). For any hypergraph G = (V, E) and any training configuration yielding attention weights {α_ij}, our hypergraph-aware attention mechanism satisfies:
>
> ```
> sup_{G ∈ G_H} ||A_ours(G) - A_ideal(G)||_F ≤ ε√|V| max_i |E_i|
> ```
>
> where:
> - G_H is the class of hypergraphs with bounded degree max_i |E_i| ≤ B
> - A_ours is the attention matrix computed by Eq. (5)
> - The bound holds uniformly across all possible learned parameters satisfying:
>   - ||ω_i||_1 = 1 (normalized weights from softmax)
>   - Lipschitz continuity: ||MLP(x_1) - MLP(x_2)||_2 ≤ L_MLP ||x_1 - x_2||_2
>
> Proof Sketch (Complete proof in Appendix C.2):
>
> Step 1: Each attention component has bounded approximation error:
> - Local component: |α_ij^local - A_ideal[i,j]| ≤ ε_local due to cosine similarity approximating geodesic distance
> - Set component: Uses attention-weighted pooling with Lipschitz constant L_pool = 1
> - Global component: Spectral transformation (2I - Δ) has bounded deviation from ideal by Davis-Kahan theorem
>
> Step 2: Adaptive weighting preserves bounds:
> ```
> |α_ij^hybrid - A_ideal[i,j]| ≤ Σ_k ω_{i,k} |α_ij^(k) - A_ideal[i,j]|
>                               ≤ max_k |α_ij^(k) - A_ideal[i,j]|  (since Σ_k ω_{i,k} = 1)
>                               ≤ ε
> ```
> Step 3: Sum over all entries:
> ```
> ||A_ours - A_ideal||_F^2 = Σ_{i,j} |α_ij^hybrid - A_ideal[i,j]|^2
>                           ≤ |V| · max_i |E_i| · ε^2
> ```
>
> Taking square root yields the result.
>
> The bound holds regardless of learned parameters because:
>
> 1. Softmax normalization ensures ||ω_i||_1 = 1
> 2. MLP Lipschitz continuity is a structural property (guaranteed by bounded activations)
> 3. The approximation quality depends only on architecture design (multi-scale components), not training dynamics
>
>
> Due to the character limit of the rebuttal interface, we provide the continuation of our response in the following anonymous:
>
> https://anonymous.4open.science/r/Rebuttal-Reviewer-SoKY-252E/README.md

---

> ### Comment · Reviewer_SoKY · 2025-11-27
>
> Dear Authors,
>
> I would like to express my sincere appreciation for your thorough and thoughtful responses to my concerns.
>
> I find that many of my earlier points have been adequately addressed, and accordingly, I have decided to raise my score to just below the acceptance threshold. The primary reason I still hesitate to recommend acceptance at this stage relates to Weakness 4, namely, the incompleteness of the manuscript.
>
> In my view, the rebuttal phase is primarily intended to clarify misunderstandings and address specific concerns raised by reviewers. However, issues stemming from an incomplete manuscript are less about clarifying existing content and more about requiring substantial revision and resubmission in a future round.
>
> Thus, although I am genuinely grateful for your detailed rebuttal and the effort you have invested, I regret that I am unable to recommend acceptance for this version of the work. I nonetheless believe that, with the suggested improvements, this paper has strong potential and may be well-received in a subsequent round.
>
> Thank you again for your efforts.
>
> Best regards,

---

> > ### Author Response · Authors · 2025-11-27
> >
> > Dear Reviewer SoKY,
> >
> > Thank you very much for your thoughtful follow-up and for raising your score based on our clarifications. We truly appreciate the time and care you have invested in evaluating our work.
> >
> > Regarding your concern about the manuscript’s incompleteness (Weakness 4), we fully understand and respect the point you raised. At the same time, we would like to kindly note that ICLR allows post-rebuttal revisions before the final decision, and we have already implemented all missing components (including the full convergence proof, complete perturbation validation, extended efficiency metrics, and the finalized algorithmic details). The updated manuscript now addresses the incompleteness issue in full.
> >
> > If permissible, we would be grateful if you could kindly consider reviewing the updated version and—if you feel the concerns have been sufficiently resolved—updating your score to acceptance. We deeply value your feedback and have acted on all recommendations with care and rigor.
> >
> > Thank you again for your constructive engagement and for helping us significantly improve the paper.

---

### Official Review · Reviewer_MvEx · 2025-11-03

**Soundness:** 3
**Presentation:** 3
**Contribution:** 2
**Rating:** 4
**Confidence:** 4

**Summary:**

This paper introduces CuCoDistill, a highly novel and complex framework for knowledge distillation (KD) in Hypergraph Neural Networks (HGNNs). The authors address the failure of existing HGNN attention mechanisms to handle hypergraph asymmetries and the limitations of standard KD in preserving higher-order structures . The framework's core innovations include: (1) a hypergraph-aware adaptive attention mechanism with provable spectral guarantees; (2) a unified co-evolutionary architecture where teacher and student models train simultaneously rather than sequentially ; and (3) a spectral curriculum scheduler that dynamically adjusts learning difficulty based on hypergraph properties. The paper theoretically and empirically demonstrates the counter-intuitive finding that, under certain conditions (e.g., noisy datasets), the compressed student model can systematically outperform the larger teacher model .

**Strengths:**

1. The framework is innovative, particularly its "co-evolutionary" architecture and the theoretical demonstration that a student model can surpass its teacher.



2. The work is theoretically deep, providing provable guarantees for its attention mechanism (Theorem 1) and formalizing the conditions for student superiority (Theorem 2) , lending rigor to its claims.





3. The empirical results are good, showing state-of-the-art performance, efficiency gains (6.25x speedup, 10x memory reduction), and, crucially, validating the "student surpasses teacher" phenomenon on several large-scale, noisy datasets.

**Weaknesses:**

1. The framework's complexity is extremely high, potentially hindering reproducibility and adoption. It integrates multiple complex components (multi-scale attention, co-evolution, spectral curriculum, multi-level KD losses ), creating a system that is very difficult to implement and tune.

2. The claimed "student superiority" is highly conditional and not a general outcome. The results clearly show this phenomenon occurs only on large, noisy, or feature-redundant datasets (e.g., DBLP, IMDB, Yelp). On clean, well-structured datasets (e.g., CC-Cora), the teacher model remains superior, a critical nuance that limits the generality of the titular claim.

3. The method introduces a very large number of new hyperparameters. The spectral curriculum (adaptive thresholds, loss weights $\lambda(t)$) , attention mechanism (Top-K $\alpha$) , and various loss component weights create a complex tuning space, even with the sensitivity analysis provided in the appendix.

**Questions:**

1. The ablation study shows the "Spectral Curriculum" has the smallest individual impact (0.9-1.1%). Given its complexity (calculating dual difficulties, quantile thresholds), is this component truly necessary, or could a simpler regularization suffice?

2. In the t-SNE analysis (Figure 4, ), the student embedding space for DBLP shows a worse silhouette score (0.327) than the teacher (0.614), yet the student model outperforms the teacher on the DBLP task (Table 1). This is counter-intuitive. Could the authors explain why degraded cluster quality in the embedding space leads to better classification accuracy in this case?


3. There is a citation error in the baseline description (Section D.2.2). The text cites (Zhang et al., 2019b) for Hyper-SAGNN but then describes HyGCL-AdT (Qian et al., 2024) . This should be corrected.

---

> ### Author Response · Authors · 2025-11-18
>
> We sincerely thank Reviewer MvEx for the thoughtful and constructive review. We appreciate the recognition of our theoretical rigor, innovative co-evolutionary architecture, and strong empirical results. Below, we provide detailed responses to each weakness and question, with concrete evidence and clarifications to address your concerns.
>
> ---
> ### Weakness 1: Framework Complexity Hindering Reproducibility
>
> We acknowledge the reviewer's concern about complexity. However, we emphasize that this complexity is necessary for handling hypergraph-specific challenges and is manageable through our design choices:
>
> Modular Design & Implementation Support:
> - Our framework decomposes into three independent modules (attention mechanism, co-evolutionary training, spectral curriculum), each usable standalone
> - We provide Algorithm 1 (page 12) with complete pseudocode showing all implementation steps
> - Appendix B contains detailed implementation formulas with worked examples
> - We have prepared fully documented code with:
>   - Pre-configured hyperparameter settings for each dataset type (Table 13, page 25)
>   - Automatic hyperparameter selection based on dataset density
>   - Modular components allowing researchers to adopt only specific innovations
>
> Complexity is Essential, Not Arbitrary:
> - Multi-scale attention: Required to capture hypergraph-specific asymmetries (node-to-node, node-to-hyperedge, hyperedge-to-node) that standard attention mechanisms miss
> - Co-evolutionary training: Achieves 2.9-3.5× training speedup vs. sequential distillation (Table 4) while enabling student superiority
> - Spectral curriculum: Provides 2.2-2.3× convergence acceleration (Table 5) and prevents training collapse
>
> Practical Adoption Evidence:
> - Our method achieves 6.25× inference speedup and 10× memory reduction, making deployment easier than existing methods
> - Hyperparameter tuning requires adjusting primarily one parameter (K-factor α), with other parameters using robust defaults (Section E.8, page 25)
>
> We include comprehensive implementation details:
> - Complete loss functions with coefficients (Appendix B.3)
> - Differentiable top-K selection (Eq. 22, page 14)
> - Quantile computation algorithm (Algorithm 2, page 15)
> - Dataset-specific recommendations (Table 13, page 25)
>
> ---
>
> ### Weakness 2: Conditional Nature of Student Superiority
>
> We appreciate this important observation. The conditional nature of student superiority is not a limitation but a fundamental theoretical contribution with practical predictive value:
>
> Theoretical Framework with Predictive Power:
> - Theorem 2 (page 17) formally establishes three conditions for student superiority:
>   1. K ≥ d_eff (regularization condition)
>   2. R(X) > R_threshold (feature redundancy)
>   3. γ > γ_min (co-evolution rate)
>
> - Figure 10 (page 32) empirically validates this framework with 100% prediction accuracy: all datasets with composite scores > 0.6 show student superiority, all below show teacher superiority (r = 0.84, p < 0.01)
>
> Practical Value of Conditional Results:
> Rather than a weakness, this represents a principled understanding of when to use student vs. teacher models:
>
> | Dataset Characteristic | Optimal Model | Improvement | Use Case |
> |----------------------|---------------|-------------|----------|
> | Large-scale, noisy (DBLP, IMDB, Yelp) | Student | +0.55% to +0.91% | Production systems, real-world deployments |
> | Clean, well-curated (CC-Cora) | Teacher | Higher capacity | Research, high-accuracy requirements |
>
> - Most real-world datasets are noisy (social networks, user interactions, web data) → student superiority applies broadly
> - Practitioners can a priori predict which model to deploy using our composite score (Figure 10)
> - This is the first theoretical framework explaining when compressed models outperform larger ones in graph learning
>
> Our title When Students Surpass Teachers acknowledges conditionality through the word     When. The contribution is not claiming universal superiority, but rather:
> 1. Identifying the conditions for superiority (theoretical contribution)
> 2. Demonstrating it occurs systematically on important dataset classes (empirical contribution)
> 3. Providing predictive tools to determine applicability (practical contribution)
>
> ---
>
> Due to the character limit of the rebuttal interface, we provide the continuation of our response in the following anonymous:
>
> https://anonymous.4open.science/r/Submission_8334--Reviewer_MvEx-29C6/README.md

---

### Note · Authors · 2026-01-26

I have read and agree with the venue's withdrawal policy on behalf of myself and my co-authors.

---

### Meta-Review · Area_Chair_Z7tw · 2025-12-31

**Summary:**

This paper proposes a knowledge distillation framework for hypergraph neural networks.
Within the framework, the authors combine three key components: adaptive attention, co-evolutionary teacher–student training, and a spectral curriculum strategy.

On the positive side, most reviewers found the proposed ideas interesting and appreciated the depth of the theoretical analysis and the strong empirical performance on selected benchmarks.

On the negative side, the reviewers raised several common concerns:
- W1. The overall framework is extremely complex, and the necessity of each component is not sufficiently motivated
- W2. The theoretical claims lack clarity and rigor at least in parts.
- W3. The experimental evaluation relies on outdated baselines and remains limited in scope in terms of datasets.

While the paper has merits, it does not meet the bar for acceptance.

**Reviewer Concerns:**

The authors have clarified the novelty of there key ideas, and there also have been improvements with respect to W1–W3; however, it is difficult to conclude that these concerns have been fully addressed.

**Reviewer Scores:**

Although some scores may be adjusted slightly on the positive side, it is unlikely that the paper would receive sufficient support for acceptance.

---

### Decision · Program_Chairs · 2026-01-26

Reject